# The Microstructure in an Al–Ti Alloy Melt: The Wulff Cluster Model from a Partial Structure Factor

**Xiaohang Lin** [1,*] , **Lin Song** [1,2], **Anchen Shao** [1], **Minghao Hua** [1], **Hui Li** [1,*] and **Xuelei Tian** [1,*]

1 Key Laboratory for Liquid-Solid Structural Evolution and Processing of Materials, Ministry of Education, School of Materials Science and Engineering, Shandong University, Jinan 250061, China; songlinwork@outlook.com (L.S.); sacboyvkx@hotmail.com (A.S.); huaminghao@outlook.com (M.H.)
2 Shandong Laboratory of Yantai Advanced Materials and Green Manufacture, Yantai 264000, China
* Correspondence: lxh12345@sdu.edu.cn (X.L.); lihuilmy@sdu.edu.cn (H.L.); xueleitian@sdu.edu.cn (X.T.)

**Abstract:** In the present work, the Wulff cluster model—which has been proven to successfully describe pure metals, homogeneous alloys, and eutectic alloys—has been extended to complex binary $Al_{80}Ti_{20}$ alloys, containing intermetallic compounds. In our model, the most probable structure in metallic melts should have the shape determined by Wulff construction within the crystal structure inside, and the cluster's size could be measured by pair distribution function. For $Al_{80}Ti_{20}$ binary alloy, three different types of clusters (Al cluster, $Al_3Ti$ cluster, and Ti cluster) were proposed. Their contributions in XRD results are investigated by a comparison with the partial XRD pattern. Ti–Ti and Al–Ti partial structural factors are completely contributed by a pure Ti cluster and an $Al_3Ti$ cluster, respectively. Al–Al partial structural factor is contributed not only by a pure Al cluster but is also related to part of the $Al_3Ti$ cluster. The simulated XRD curve shows a good agreement with the experimental partial $I(\theta)$, including the peak position, width, and relative intensity.

**Keywords:** Al–Ti alloy; melt; intermetallic compound; Wulff cluster model; DFT





## 1. Introduction

Titanium (Ti)-based alloys are widely applied as a high-performance, lightweight material for marine light engineering equipment due to their low density, high specific strength, non-magnetism, and strong seawater corrosion resistance [1–4]. Compared with other materials, Al–Ti alloys owners' superior marine adaptability has attracted great attention and interest [5–7]. Considering the different application scenarios of Al–Ti alloys in a variety of complex areas, it is necessary to design its structure, composition, and physical and chemical properties to achieve more extensive applications. This puts forward higher requirements for the in-depth study of the preparation process of Al–Ti alloys, especially their initial state—a melt structure. Unfortunately, the melt structure of Ti-based alloys remains unclear.

In recent years, the exploration of the melt structure of alloys has been frequent. High-temperature X-ray diffraction (HTXRD), synchrotron radiation X-ray diffraction, and X-ray absorption fine structure (XAFS), etc., are used as common methods to directly characterize its details [8–12]. However, due to the special, high temperature, liquid environment of alloy melt, these direct characterization methods can not reflect the real microscopic physical image of the melt. In this case, it is reasonable to use a model description method [13–16], such as the crystal defect model, the quasi-crystalline mode, or the cluster model, etc. [17–24].

In our previous works, a thermodynamic model based on the Wulff theory was proposed, the so-called Wulff cluster model, which was successfully applied to describe the melt structures of pure metals and binary homogeneous and eutectic alloys [25–27]. The simulated results have shown good agreement with the experimental results, and some phenomena, such as nucleation processes, have been directly indicated by the simulation;

note that, in the Wulff cluster model, the most probable cluster structure is used to describe a structure distribution in metallic melts. Our model mainly contains three points. First, the metallic melts are treated as thermodynamic equilibrium systems, which can be proved by repeatable results, obtained from HTXRD experiments. Therefore, the shape of the cluster in the system could be described by the Wulff theory [28–30]. According to the theory, the structure and morphology of nanoparticles under the condition of thermodynamic equilibrium are determined by the surface energies of planes. Second, the clusters have the crystal structure inside. It is based on the fact that the characteristic peaks of the HTXRD intensity curve are somehow related to the solid-state XRD [26]. Third, the pair distribution function (PDF) g($r$), based on the HTXRD I($\theta$) conversion, was applied to determine the cluster's size [28,29,31].

In this work, the Wulff cluster model is used to describe Al–Ti alloy melts, including not only the pure metal cluster, but also intermetallic compounds with characteristic atomic stoichiometry and lattices. The partial structure factors from experiments are analyzed in order to investigate the internal details of the melt structure with multiple mixed phases.

## 2. Methods

### 2.1. Theoretical Methods

All the calculations in this paper are performed by the Vienna ab initio simulation package (VASP) [32,33], based on the density functional theory (DFT) [34,35]. The exchange correlation functional was described through the generalized gradient approximation (GGA) with the Perdew, Burke, and Ernzerhof (PBE) parameter, which has already been proved to give a reasonable description of metal systems [36,37]. The ionic cores were represented by projector augmented wave (PAW) potentials [38,39]. Based on a precise convergence test, the value of the cut-off energy was set as 400 eV for plane wave expansions in reciprocal space. Energy calculations of surface models were performed in the first Brillouin zone, using the $11 \times 11 \times 1$ k-points in the Monkhorst–Pack scheme to confirm a good convergence of the total energy. For the control of the nuclear motion convergence accuracy, the value of the energy convergence accuracy of our system was $1.0 \times 10^{-4}$ eV.

The implicit solvation model (VASPsol) is used to calculate the solid–liquid interface energy by treating the liquid solvent as an average field, which was proven to be a reasonable model to approximately describe the solid–liquid interface [40,41]. In order to determine the key parameters (solvent dielectric constant (EB_k)) of the melt structure, calculated by the implicit solvation model, a large range of calculations have been carried out in previous works [27]. Although the values of interface energies are different, the ratios of interface energies are similar to the ones of surface energies. In this case, it is reasonable to use the surface energy to describe the structure of the Wulff model instead of the interface energy.

A typical two-sided model was proposed to calculate the surface energies of different crystal planes. The model was constructed by selectively exposing specific planes with a large enough vacuum (25 Å). All slab models were constrained to the symmetric top and bottom surfaces. For pure metals, the surface energy $\gamma$ is determined by the following formula:

$$\gamma = 1/2A(E_{slab} - NE\_bulk) \tag{1}$$

where $A$ means the total area of the facet, $E\_slab$ represents the generated energy of the generated slab model, $E\_bulk$ is the energy of the bulk unit cell, and $N$ is the number of atoms in bulk structure.

For intermetallic compounds, the surface structure is more complex than that of pure metals, which is reflected in the different stoichiometric ratio and several different layers in the same crystal plane family. Therefore, the surface energy properties can be better described by the following formula, with chemical potential:

$$\gamma = 1/(2A)[2E\_surf - \mu_{XY}N_Y - \mu_X(N_Y - N_X)] \tag{2}$$

where $A$ is the area of intermetallic surface, $E_{surf}$ is the total energy of the intermetallic surface per surface unit cell, $\mu_{XY}$ and $\mu_X$ are the chemical potential of intermetallic compounds and pure metal, and $N_X$ and $N_Y$ are the atomic numbers of the $X$ and $Y$ elements of intermetallic compounds, respectively.

After careful calculation and verification, the surface free energy does not change under a limited temperature (compared with 0 K, the surface free energy difference is less than 0.5% at 800 K), which agrees with the results of other researchers [42]. It indicates that the temperature effect should hardly influence the surface free energy, so that the surface energy is used instead of free energy in our calculation.

In order to verify the rationality of the structure of Al–Ti melts, simulated XRD would be directly compared with the experimental diffraction peak, which needs to complete two key steps: 1. Obtain the XRD patterns of simulation determined cluster structure. The powder diffraction module of the Materials Studio software was applied to calculations, which were based on Rietveld refinement, Rietveld with energies, Pareto optimization, and modified Pawley refinement [43–46]. 2. Broaden solid-state simulated XRD patterns. Considering the small size and lattice distortion caused by the temperature effect, the broadening formula was presented in the form of normal distribution as follows [47]:

$$I(2\theta) = \sum_{i=1}^{n} \left\{ I_i \left[ \frac{P_1}{\sqrt{a_1^2 + b^2}\sqrt{2\pi}} e^{-\frac{(2\theta - 2\theta_i)^2}{2(a_1^2 + b^2)}} + \frac{P_2}{\sqrt{a_2^2 + b^2}\sqrt{2\pi}} e^{-\frac{(2\theta - 2\theta_i)^2}{2(a_2^2 + b^2)}} \right] \right\} + G \left[ 1 - D^2(s) \right] \quad (3)$$

where $I_i$ means the intensity of the number $i$ XRD peak of a crystal lattice, and $P_1$ and $P_2$ are the ratio of the amount of the atoms in the inner part and the surface part of the atomic short-range ordering, respectively. The $a_1$ and $a_2$ express the coefficients that indicate the $I(2\theta)$ breadth, and b represents a coefficient that is related to the breadth of the broadening peak. The XRD angle is indicated by $2\theta$ and $2\theta_i$ is the position of the peak $i$ of a crystal lattice. In $D^2(s) = e^{-Bs^2/2}$, B is a temperature coefficient present the effect of the thermal vibration, $s = 2\sin\theta/\lambda$, and G is the coefficient of the background. For metals, the above parameters are basically determined by experiments, except for temperature, other parameters are constant.

The structure factor $S(Q)$ can be obtained from the diffraction intensity $I(\theta)$, which depends on the HTXRD experiment after treating by polarization, absorption correction, and normalization correction. The $S(Q)$ also can be transformed into a pair distribution function $g(r)$ by Fourier transform, as shown in Equation (4):

$$g(r) = 1 + \frac{1}{2\pi r^2 \rho_0} \int_0^\infty Q[S(Q) - 1]\sin Qr dQ \quad (4)$$

where $Q = 4\pi\sin\theta/\lambda$ and $\rho_0$ is the number density of the metal at a certain temperature. In the binary system, it can be described by partial structure factor $S_{ij}(Q)$ and partial pair distribution function $g_{ij}(r)$, which is determined by calculating the average weighting function of atoms, which satisfy the following formula:

$$S_{ij}(Q) = \omega_{ii}S_{ii}(Q) + \omega_{jj}S_{jj}(Q) + 2\omega_{ij}S_{ij}(Q) \quad (5)$$

$$g_{ij}(r) = \omega_{ii}g_{ii}(r) + \omega_{jj}g_{jj}(r) + 2\omega_{ij}g_{ij}(r) \quad (6)$$

$$\omega_{ij} = c_i c_j f_i f_j / \langle f \rangle^2 \quad (7)$$

where $S_{ii}(Q)$, $S_{jj}(Q)$, and $S_{ij}(Q)$ are the structure factors of *i-i*, *j-j*, and *i-j* type atom pairs, respectively. $c_i$ means the atomic fraction of *i*-type atoms and $f_i$ is the atomic scattering factor of *i*-type atoms.

*2.2. Experimental Methods*

The $Al_{80}Ti_{20}$ alloys were converted into mass ratio and prepared. The 99.999% high-purity of materials (Al and Ti) were melted into the required alloys in vacuum melting (Physcience Opto-electronics Co., Ltd., Beijing, China) and were treated as samples for HTXRD experiments. The surface of heated high-temperature melt was irradiated by $K\alpha$ ray (wavelength $\lambda = 0.07089$ nm), excited by Mo target, and the diffracted beam reached the detector (Bruker Daltonics, Leipzig, Germany) through the graphite detector (Bruker Daltonics, Leipzig, Germany). The whole diffraction experiments were carried out in a high purity helium (99.999%) atmosphere ($1.3 \times 10^5$ Pa). The alloy samples were placed in an alumina crucible with a size of 30 mm $\times$ 25 mm $\times$ 8 mm and heated with tantalum plates. After being heated to 1500 °C, they were held for 1 h, and cooled down to the required temperature. The X-ray scanning voltage is 40 kV, the current is 30 mA, the exposure time is 30 s, and the measurement angle (2θ) is 5° to 80°.

## 3. Results and Discussion

Although the Wulff cluster model could successfully describe the melts structure of pure metals and binary homogeneous and eutectic alloys [25–27], one should be very careful to extend it to general binary alloys containing intermetallic compounds. This is because there are several possible cluster types in such melts, which will make the situation complex. Moreover, the melt structures cannot be fully reflected by the solid structures after solidification. Fortunately, partial structural factors S($Q$) have built a "bridge" to study such binary systems.

As shown in Figure 1a, there are several intermetallic compounds with completely different lattice structures in the whole composition range of the Al–Ti phase diagram. Because of the relatively simple composition and low melting temperature, $Al_{80}Ti_{20}$ was applied to explore the binary Al–Ti system. The cluster types that might exist in $Al_{80}Ti_{20}$ melts include pure Al (FCC), pure Ti (BCC at high temperature), and $Al_3Ti$ (SC) lattice, shown in Figure 1b–d, respectively. In $Al_3Ti$ crystal (Figure 1c), every Ti atom is surrounded by six Al atoms with the distance of 2.8 Å. Although the Ti–Ti distance is not so large (about 4.0 Å), there is no family or plane formed only by Ti atoms. In this case, the $Al_3Ti$ cluster should not contribute to Ti–Ti partial structural factors; however, we obtained Ti–Ti partial structural factors in the $Al_{80}Ti_{20}$ melt. The only explanation is there are clusters of pure Ti metal in this melt. According to the phase diagram, clusters of pure Al should exist in $Al_{80}Ti_{20}$ melt. Moreover, $Al_3Ti$ clusters also contribute to the Al–Al partial structural factors, since some of the crystal planes in the $Al_3Ti$ crystal structure were formed only by Al atoms. It is obvious that the $Al_3Ti$ cluster was the only one which contributed to Al–Ti partial structural factors.

The results above are totally based on the Wulff cluster model and the crystal structure, and we discuss and proof separately, by comparing the simulated XRD pattern with the three partial structural factors. At the first step, the atomic structure of the three types of clusters should be determined. As we mentioned in the introduction, according to our Wulff cluster model, the most probable structure in metallic melts should have the shape determined by Wulff construction, within the crystal structure inside, and the cluster's size could be measured by the pair distribution function in the experiments.

Furthermore, there is the following question: how can one determine the specific size of different types of clusters for this complex binary alloy system? The average size of the melt cluster was acquired, in which g(r) = 1 ± 0.02 was generally considered as the range of short-range ordering. The corresponding partial S($Q$) curve was obtained after polarization, correction, and normalization, which showed a good agreement with previous others [25–27], represented in Figure 2a. On the whole, the partial S($Q$) curves are smooth and two apparent peaks appeared. It can be seen that there was a certain deviation in the intensity and position of the partial S($Q$) peak, describing the interaction between different elements (Al–Al, Al–Ti, and Ti–Ti), especially the positions of first and

second peaks. The partial PDF g(r) of the $Al_{80}Ti_{20}$ alloy melt can be obtained by a Fourier transform from the partial S(Q) (Equation (4)).

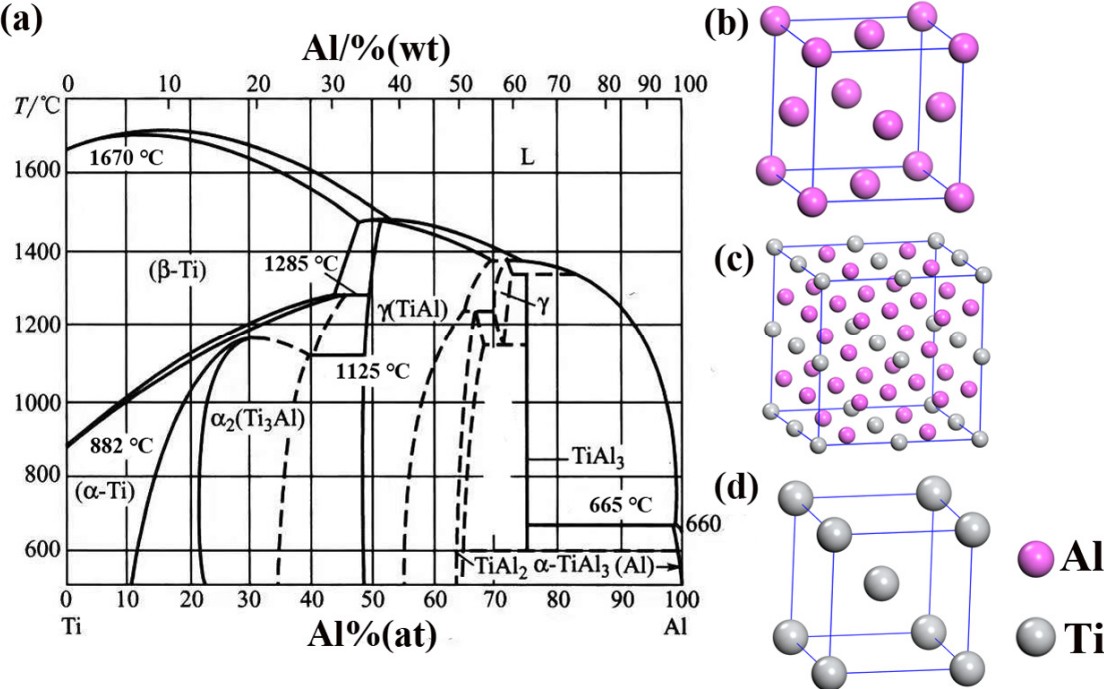

**Figure 1.** (**a**) The phase diagram of Al–Ti alloys; (**b**) the unit cell of Al FCC bulk; (**c**) the $2 \times 2 \times 2$ supercell of $Al_3Ti$ intermetallic; (**d**) the unit cell of Ti BCC bulk (high temperature).

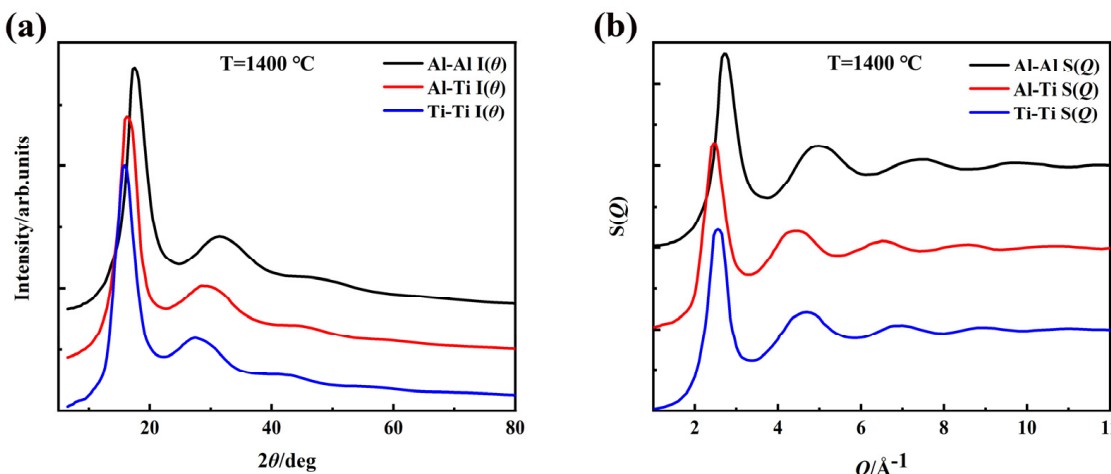

**Figure 2.** The $Al_{80}Ti_{20}$ alloy melt. (**a**) HTXRD partial diffraction intensity S(Q); (**b**) partial pair distribution function g(r). Here, black, red, and blue lines represent the partial S(Q) or g(r) of Al–Al, Al–Ti, and Ti–Ti, respectively.

Three curves in Figure 2b show that the relevant radiuses are 9.73 Å (Al–Al), 8.26 Å (Al–Ti) and 8.47 Å (Ti–Ti), respectively. Note that $Al_3Ti$ clusters contribute not only to the Al–Ti but also the Al–Al partial structural factors. It is reasonable to use the average value of relevant radiuses from these two partial structural factors as the relevant radiuses for the size of $Al_3Ti$ clusters. The size of the Al and Ti clusters should be correctly described by the relevant radiuses from the Al–Al and Ti–Ti partial structural factors.

In order to determine the Wulff construction of three types of clusters (Al, Ti, and $Al_3Ti$ clusters), the surface energy in different crystal planes for three crystals are calculated, which is shown in Table 1. For the BCC Ti crystal, the Ti(100) surface is the closed packed

surface that has the lowest surface energy. For the FCC Al, there exists the order of $\gamma$ (111) < $\gamma$ (100) < $\gamma$ (110) in low-index surfaces. For the Al$_3$Ti crystal, most of the families of crystal plane contain several different surfaces as shown in Table 1. Recall that, for non-stoichiometric surfaces, the environment contributes to the stability of the surfaces through the corresponding chemical potentials (shown in Equation (2)). Normally, the surface energy can be described as a function of chemical potential. To focus on our key point, a special chemical potential that is equal to the chemical potential of bulk Ti is used to make it easy to understand. In fact, most of the Al$_3$Ti surfaces that we considered are not stoichiometric. As we are using slab models to calculate surface energies, there are always two surfaces present in our surface model, which are not necessarily equal. However, typically, the different surface energies for asymmetric slabs cannot be separated, so that only an average surface energy can be derived. In the case of the Al$_3$Ti surfaces, we were able to always construct slabs with the same first layer of atoms, but the second layer can differ. Despite this, we take the average value, which is reasonable considering that the first layer on both sides is the same.

**Table 1.** The surface energies of various crystal planes for Al, Ti, and Al$_3$Ti crystals.

| Types | $\gamma$ (J/m$^2$) | Types | $\gamma$ (J/m$^2$) | Types | $\gamma$ (J/m$^2$) |
|-------|-------|-------|-------|-------|-------|
| Ti(100) | 1.56 | Al(100) | 0.96 | Al$_3$Ti (100)-1 | 1.85 |
| Ti(110) | 1.73 | Al(110) | 1.03 | Al$_3$Ti (100)-2 | 1.23 |
| Ti(111) | 1.82 | Al(111) | 0.87 | Al$_3$Ti (110)-1 | 1.93 |
| Ti(210) | 1.69 | Al(210) | 1.03 | Al$_3$Ti (110)-2 | 1.15 |
| Ti(211) | 1.73 | Al(211) | 0.96 | Al$_3$Ti (111) | 1.09 |
| Ti(221) | 1.69 | Al(221) | 0.96 | Al$_3$Ti (210)-1 | 1.93 |
| Ti(310) | 1.72 | Al(310) | 1.03 | Al$_3$Ti (210)-2 | 1.91 |
| Ti(311) | 1.81 | Al(311) | 0.99 | Al$_3$Ti (211)-1 | 1.56 |
| Ti(511) | 1.78 | Al(511) | 0.98 | Al$_3$Ti (211)-2 | 1.28 |
|  |  |  |  | Al$_3$Ti (221) | 1.21 |

The Wulff shapes of Ti, Al$_3$Ti, and Al were constructed according to the calculated surface energies, as shown in Figure 3a. Since the crystal structures of the three materials are totally different (Ti (BCC), Al$_3$Ti (SC), and Al (FCC)), the Wulff constructions are different not only in the number of exposed faces, but also the type and proportion of the surfaces. Clearly, the surfaces with largest areas of all the Wulff shapes are the closed packed surfaces.

Here, the question of whether the atomic cluster structures of Al, Al$_3$Ti, and Ti can accurately describe the melt structure characteristics of Al80Ti20 alloy arises. Next, the partial XRD pattern was compared with the simulated one by corresponding cluster. Recall that, from our analysis, all Ti–Ti partial structural factors should come from pure Ti clusters in the melt. In order to verify the rationality of this conjecture, the diffraction intensity curve of the Ti cluster, simulated by the Materials Studio, was directly compared with the experimental partial Ti–Ti diffraction intensity after broadening (Equation (3)), as shown in Figure 4. The simulated XRD diffraction curves of Ti clusters are in good agreement with the partial Ti–Ti XRD diffraction intensity, including the position and relative intensity of the first and second peaks. This is strong evidence that, not only the clusters corresponding to the phase shown in the phase diagram, but also some other clusters with strong inner interaction, could exist (similarly to the Ti cluster).

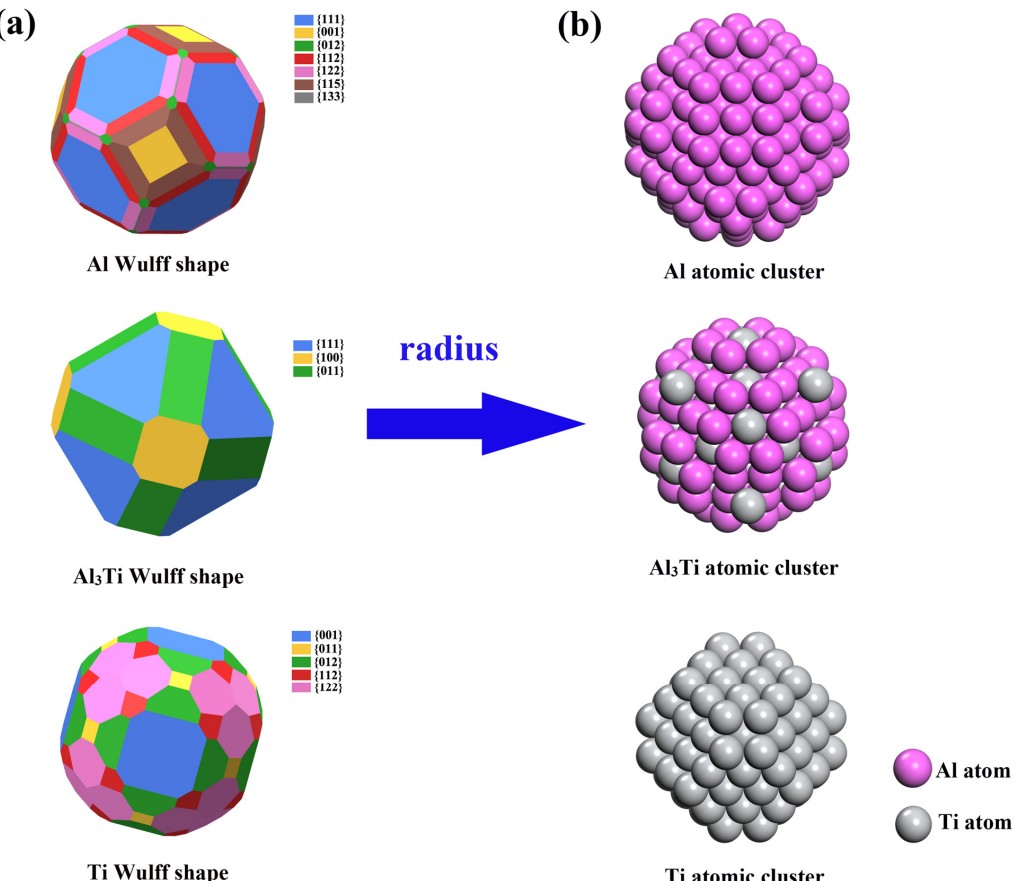

**Figure 3.** (**a**) The Wulff shape of Al, Al$_3$Ti, and Ti (BCC). (**b**) The atomic cluster-determined radius of Al, Al$_3$Ti, and Ti (BCC).

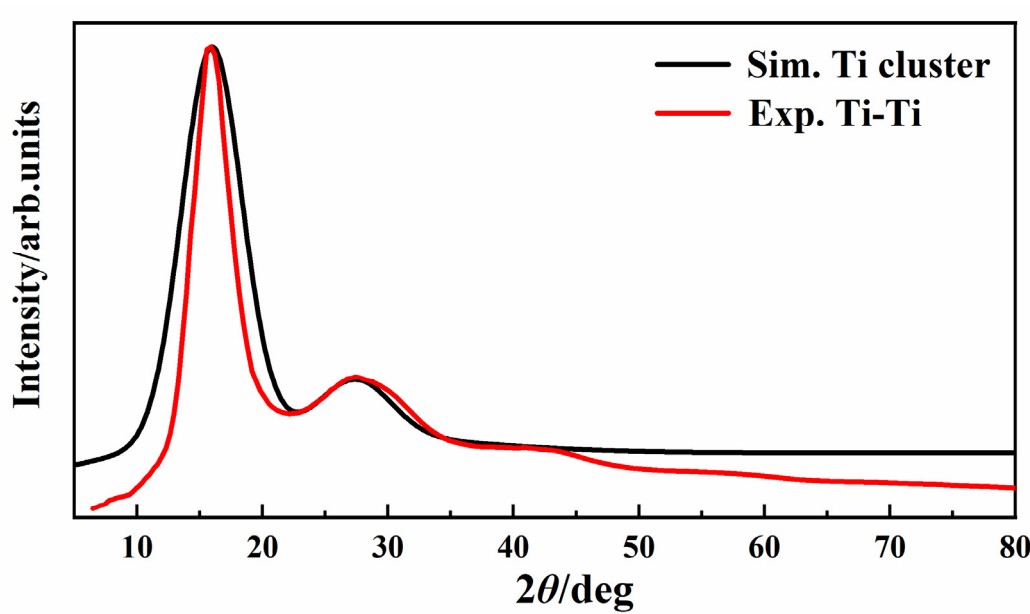

**Figure 4.** Comparison of the experimental partial Ti–Ti XRD pattern with the simulated Ti cluster.

As mentioned above, Al–Al partial structural factor should be contributed to, not only by the pure Al cluster, but also by the related part of Al$_3$Ti. To prove the inference, one should combine the simulated XRD pattern of the pure Al cluster and the related part of Al$_3$Ti, as the simulated partial Al–Al XRD pattern. As predicted, if one just compares

the experimental partial Al–Al HTXRD curve (red line in Figure 5) and the simulated one of pure Al cluster (blue dashed line in Figure 5), there is a large deviation—neither the intensity nor the position of the second peak agree with each other. It is challenging to separate partial Al–Al XRD from the simulated Al$_3$Ti cluster XRD pattern. In our research, the XRD pattern of Al$_3$Ti cluster structure, without Ti atoms (yellow dashed line in Figure 5), is approximately treated as the partial Al–Al XRD. The XRD pattern that we obtained in this manner includes part of the Al–Ti structure. At least, it contains all information of the partial Al–Al XRD of the Al$_3$Ti cluster. Fortunately, we obtained rather positive result. The combination XRD curve of the pure Al cluster and the partial Al–Al XRD in the Al$_3$Ti cluster (black curve in Figure 5) show good agreement with the experimental result, especially the relative intensity of the first and second peaks. Although the position of the second peak deviates a little bit, this is mainly caused by the approximate treatment of the Al–Al XRD in the Al$_3$Ti cluster, as mentioned above.

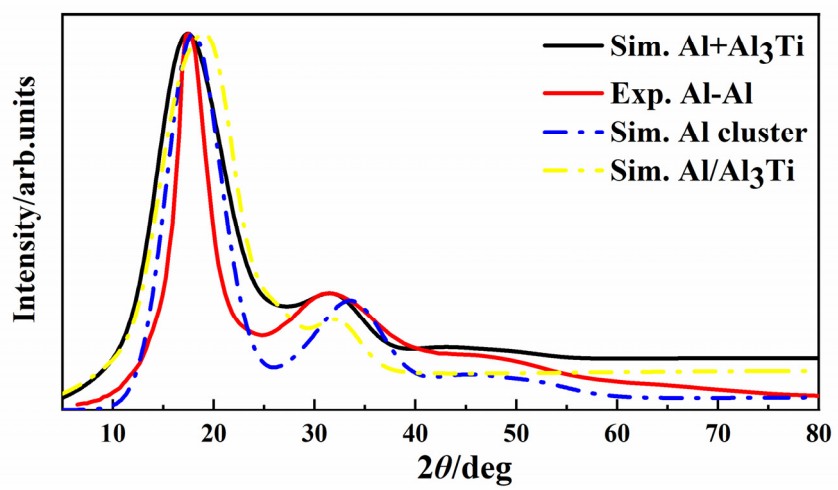

**Figure 5.** Comparison of the experimental partial Al–Al XRD pattern with simulated one.

As has been established the Al–Ti partial structural factor is completely contributed by the Al$_3$Ti cluster; however, the Al$_3$Ti cluster contributes not only to the Al–Ti, but also to the Al–Al partial structural factors. The Al–Ti partial XRD pattern and the simulated one of Al$_3$Ti cluster (shown in Figure 6a) are not comparable; however, it is difficult to divide the XRD pattern of the Al$_3$Ti cluster into the Al–Ti and Al–Al parts. Considering that the Al–Al partial structural factor is contributed to by both the Al and Al$_3$Ti clusters, it is reasonable to compare the combination of the Al–Al and Al–Ti partial XRD patterns (red line in Figure 6b) with the combination of the simulated XRD results of the pure Al and Al$_3$Ti clusters. To determine the composition ratios of Al clusters and Al$_3$Ti clusters, the simulated diffraction peaks, formed by different composition ratios (100% Al cluster; 60% Al, 40% Al$_3$Ti; 50% Al, 50% Al$_3$Ti; 40% Al, 60% Al$_3$Ti; 100%Al$_3$Ti cluster) were shown in Figure 6b (black line, yellow line, blue line, green line, and pink dashed line, respectively). The composition of 50% Al and 50% Al$_3$Ti simulated curve showed an excellent agreement with the experimental curve. In summary, the Wulff cluster model can successfully describe the melt structure of Al–Ti alloys and can explain the contribution of every kind of clusters. The most important point is that the atomic structure of the cluster we got is not a cluster of specific existence in melts. It is the most probable cluster structure of a structure distribution in melts, through the statistical average of time and space, similarly to X-ray diffraction.

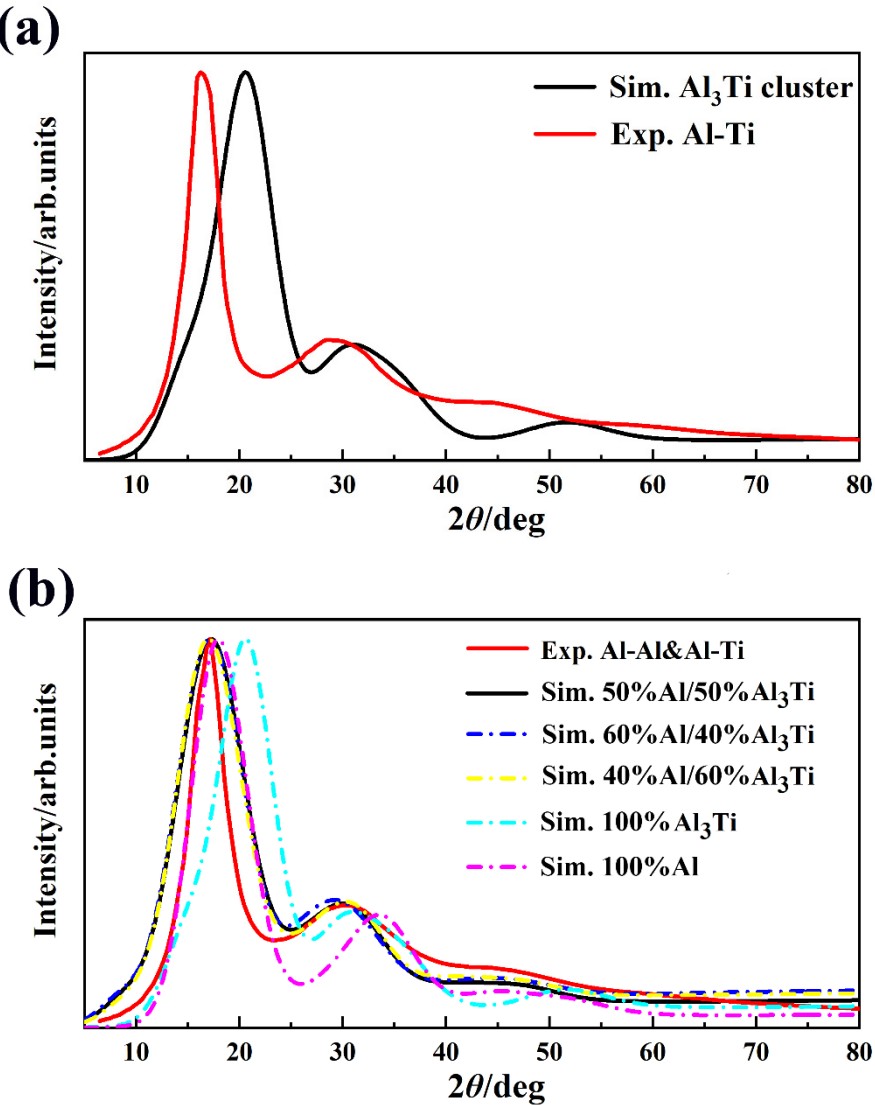

**Figure 6.** Comparison of the combination of Al–Al and Al–Ti partial XRD pattersn (red line) with simulated lines of: (**a**) Al$_3$Ti cluster; (**b**) combination of simulated XRD result of pure Al and Al3Ti cluster with different composition ratios.

## 4. Conclusions

In this work, the Wulff cluster model was applied to a complex binary Al$_{80}$Ti$_{20}$ alloy containing intermetallic compounds. For this model, the shape was determined by the surface energies based on DFT calculations within the crystal structure inside, and the sizes were obtained from the PDF g(*r*), converted from the HTXRD experimental results. For the Al$_{80}$Ti$_{20}$ binary alloy system, three different types of clusters (Al cluster, Al$_3$Ti cluster, and Ti cluster) were proposed, and their contributions in the XRD results were investigated through a comparison with the partial XRD pattern. The simulated XRD diffraction curves of Ti clusters are in good agreement with the partial Ti–Ti XRD diffraction intensity. This is strong evidence that, not only the clusters corresponding to the phase shown in the phase diagram, but also some other clusters with strong inner interaction, could exist (similarly to the Ti cluster). Similarly, the Al–Ti partial structural factors were completely contributed by the Al$_3$Ti cluster, respectively. Note that the Al–Al partial structural factor is contributed not only by the pure Al cluster, but also by the related part of the Al$_3$Ti cluster. The combination XRD curve simulated by both the Al$_3$Ti cluster and the pure Al cluster completely agreed with the experimental partial I(θ), including the peak position, width, and relative intensity. The Wulff cluster model can successfully describe the melt structure of Al–Ti alloys and

can explain the contribution of every kind of cluster. The most important point is that the atomic structure of the cluster we obtained is not a cluster of specific existence in melts. It is the most probable cluster structure of a structure distribution in melts, through the statistical average of time and space, similarly to X-ray diffraction.

**Author Contributions:** Conceptualization, X.L., H.L. and X.T.; methodology, L.S.; software, X.L.; validation, A.S. and M.H.; formal analysis, X.L.; investigation, L.S.; resources, L.S.; data curation, X.L.; writing—original draft preparation, L.S.; writing—review and editing, X.L.; visualization, A.S. and M.H.; supervision, X.L.; project administration, H.L. and X.T.; funding acquisition, X.T. All authors have read and agreed to the published version of the manuscript.

**Funding:** This research was funded by [the National Natural Science Foundation of China] grant number [U1806219] and [China Postdoctoral Science Foundation] grant number [2018M642642].

**Data Availability Statement:** Not applicable.

**Acknowledgments:** X.L. and L.S. contributed to the work equally and should be regarded as co-authors. We gratefully acknowledge the financial support from the National Natural Science Foundation of China (Project No. U1806219), China Postdoctoral Science Foundation (Project No. 2018M642642).

**Conflicts of Interest:** The authors declare no conflict of interest.

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
