# Peer review of "The Microstructure in an Al–Ti Alloy Melt: The Wulff Cluster Model from a Partial Structure Factor"

_metals, doi:10.3390/met11111799_

Round 1
Reviewer 1 Report
I liked the work, in the introduction you have a good overview of the literature. There is also literature from 2021. Drawings are made with high quality, informative, readable. It's good that we compared simulations with experimental data.

Author Response
I liked the work, in the introduction you have a good overview of the literature. There is also literature from 2021. Drawings are made with high quality, informative, readable. It's good that we compared simulations with experimental data.
Response: We appreciate that the reviewer emphasizes that we address “have a good overview of the literature; Drawings are made with high quality, informative, readable; It's good that we compared simulations with experimental data.”.
Reviewer 2 Report
Strengths
In this work, the Wulff cluster model was applied to complex binary Al80Ti20 alloy containing intermetallic compound. For this model, the shape was determined by the surface energies based on DFT calculations within the crystal structure inside, and the sizes were obtained from the PDF g(r) converted from the HTXRD experimental results. For Al80Ti20 binary alloy system, three different types clusters (Al cluster, Al3Ti cluster and Ti cluster) were proposed and their contributions in XRD results are investigated by comparing with the partial XRD pattern. Ti-Ti and Al-Ti partial structural factors are completely contributed by pure Ti cluster and Al3Ti cluster respectively. Al-Al partial structural factor is contributed not only by pure Al cluster but also related part of Al3Ti cluster. The simulated XRD curve showed a good agreement with the experimental partial I(θ), including the peak position, width and relative intensity.
It is necessary to note the original approach of the authors in measuring the intensity of HTXRD S(Q) of partial diffraction the binary Al80Ti20 alloy.
Weakness
- To test the Wolfe cluster model, it would be advisable to calculate the size of the clusters based on the activation energy of the viscous flow. Activation energy is obtained from the analysis of temperature dependencies of heating and cooling of melt kinematic viscosity.
- The references need to be updated.
- The formatting of the references should be brought into line with the requirements of the Journal.

Author Response
Strengths
In this work, the Wulff cluster model was applied to complex binary Al80Ti20 alloy containing intermetallic compound. For this model, the shape was determined by the surface energies based on DFT calculations within the crystal structure inside, and the sizes were obtained from the PDF g(r) converted from the HTXRD experimental results. For Al80Ti20 binary alloy system, three different types clusters (Al cluster, Al3Ti cluster and Ti cluster) were proposed and their contributions in XRD results are investigated by comparing with the partial XRD pattern. Ti-Ti and Al-Ti partial structural factors are completely contributed by pure Ti cluster and Al3Ti cluster respectively. Al-Al partial structural factor is contributed not only by pure Al cluster but also related part of Al3Ti cluster. The simulated XRD curve showed a good agreement with the experimental partial I(θ), including the peak position, width and relative intensity.
Response: We thank the reviewer for carefully reading our manuscript and given some rather constructive comments.
It is necessary to note the original approach of the authors in measuring the intensity of HTXRD S(Q) of partial diffraction the binary Al80Ti20 alloy.
Response:
We are sorry that we did not mention this part in the experimental method. The following part has been added into the method section:
“2.2 Experimental methods
The Al80Ti20 alloy were converted into mass ratio and were prepared. The 99.999% high-purity of materials (Al and Ti) are melted into the required alloys in vacuum melting, which treated as samples for HTXRD experiments. The surface of heated high-temperature melt was irradiated by Kα ray (wavelength λ = 0.07089 nm) excited by Mo target and the diffracted beam reached the detector through the graphite detector. The whole diffraction experiments were carried out in high purity helium (99.999%) atmosphere (1.3 × 105 Pa). The alloy samples were placed in an alumina crucible with a size of 30 mm× 25 mm × 8 mm and heated with tantalum plates. After heating to 1500℃, held for 1 hour and cooled down to the required temperature. The X-ray scanning voltage is 40 kV, the current is 30 mA, the exposure time is 30 s, and the measurement angle (2θ) is 5° to 80°.”
Weakness
To test the Wolfe cluster model, it would be advisable to calculate the size of the clusters based on the activation energy of the viscous flow. Activation energy is obtained from the analysis of temperature dependencies of heating and cooling of melt kinematic viscosity.
Response: Thank you so much for review’s suggestion. It is well to know such method which can test our model. We will focus on this experiment in the next stage of our research.
The references need to be updated.
Response: Thanks for reviewer’s suggestion. The following references have been updated or added:
- “33. Harrison, M. J.; Woodruff, D. P.; Robinson, J.; Sander, D.; Pan, W.; Kirschner, J., Adsorbate-induced surface reconstruction and surface-stress changes in Cu (100)∕ O: Experiment and theory. Physical Review B 2006, 74, (16), 165402.
- 34. Bonzel, H. P.; Yu, D. K.; Scheffler, M., The three-dimensional equilibrium crystal shape of Pb: Recent results of theory and experiment. Applied Physics A 2007, 87, (3), 391-397.
- 45. Hacene, M.; Anciaux Sedrakian, A.; Rozanska, X.; Klahr, D.; Guignon, T.; Fleurat Lessard, P., Accelerating VASP electronic structure calculations using graphic processing units. Journal of computational chemistry 2012, 33, (32), 2581-2589.
- 46. Hutchinson, M.; Widom, M., VASP on a GPU: Application to exact-exchange calculations of the stability of elemental boron. Computer Physics Communications 2012, 183, (7), 1422-1426.
- 47. Einstein, T. L., Equilibrium shape of crystals. In Handbook of Crystal Growth, Elsevier: 2015; pp 215-264.”
The formatting of the references should be brought into line with the requirements of the Journal.
Response: We are sorry for our mistake. The format of the references has been modified.
Special thanks to you for your good comments.
Reviewer 3 Report
Formula (3) should be written in one line, and if it cannot fit, it should be formatted appropriately.
The conclusion should be slightly expanded and more precise explanations of the research results should be given.
Author Response
Formula (3) should be written in one line, and if it cannot fit, it should be formatted appropriately.
Response: Thanks for the reviewer’s suggestion. We have modified formula 3 and now it is in one line.
The conclusion should be slightly expanded and more precise explanations of the research results should be given.
Response: Thanks for the reviewer’s suggestion. We slightly expanded the conclusion as following:
“In this work, the Wulff cluster model was applied to complex binary Al80Ti20 alloy containing intermetallic compound. For this model, the shape was determined by the surface energies based on DFT calculations within the crystal structure inside, and the sizes were obtained from the PDF g(r) converted from the HTXRD experimental results. For Al80Ti20 binary alloy system, three different types clusters (Al cluster, Al3Ti cluster and Ti cluster) were proposed and their contributions in XRD results are investigated by comparing with the partial XRD pattern. The simulated XRD diffraction curves of Ti clusters are in good agreement with the partial Ti-Ti XRD diffraction intensity. This is strong evidence that not only the clusters corresponding to the phase shown in the phase diagram, but also some other clusters with strong inner interaction could exist (like Ti cluster). Similarly, Al-Ti partial structural factors are completely contributed by Al3Ti cluster respectively. Note that, Al-Al partial structural factor is contributed not only by pure Al cluster but also related part of Al3Ti cluster. The combination XRD curve simulated by both Al3Ti cluster and pure Al cluster completely agree with the experimental partial I(θ), including the peak position, width and relative intensity. Wulff cluster model can well describe the melt structure of Al-Ti alloy and can explain the contribution of every kind of clusters. The most important point is that the atomic structure of cluster we got is not a cluster of specific existence in melts. It is the most probable cluster structure of a structure distribution in melts through the statistical average of time and space, just like what Xray diffraction does.”
Special thanks to you for your good comments.